# Bio-Assay-Guided Isolation of Fractions and Constituents with Antioxidant and Lipid-lowering Activity from *Allium cepa*

**DOI:** 10.3390/antiox12071448

**Published:** 2023-07-18

**Authors:** Kongming Fan, Yan Li, Qiaofeng Fu, Jinmin Wang, Yong Lin, Linyu Qiu, Li Ran, Junli Yang, Chunyan Yang

**Affiliations:** 1School of Pharmacy, Institute of Material Medica, North Sichuan Medical College, Nanchong 637100, China; a765358710@163.com (K.F.); ly1369968@163.com (Y.L.); 13882946025@163.com (Q.F.); 18212321154@163.com (J.W.); 17390215317@163.com (L.Q.); 2Innovative Platform of Basic Medical Sciences, School of Basic Medical Sciences, North Sichuan Medical College, Nanchong 637100, China; 3School of Public Health, North Sichuan Medical College, Nanchong 637100, China; linyong@nsmc.edu.cn; 4Lanzhou Institute of Chemical Physics, Chinese Academy of Sciences, Lanzhou 730000, China

**Keywords:** *Allium cepa*, lipid lowering, antioxidant, effective fractions, active constituents

## Abstract

Active fractions and constituents with antioxidant and lipid-lowering activities were investigated using bio-assay-guided isolation and identification. The data showed that the antioxidant fraction of *A. cepa* was AC50%, the main constituents of which were quercetin and isoquercitrin, by way of both ultra-high performance liquid chromatography–mass spectrometry (UPLC-MS) and bio-assay-guided purification and elucidation. Similarly, the lipid-lowering active fraction of *A. cepa* was AC30% with the main constituents of 3,4-dihydroxybenzoic acid and quercetin 3,4′-*O*-diglucoside. Also, bio-assay-guided isolation led to the isolation and identification of five known compounds with a purity of more than 98%, and quercetin was both the best free radical scavenger and lipid-lowering constituent. Moreover, the mechanism of the lipid-lowering effect of AC30% might be its reduction in mRNA expression levels of sterol regulatory element binding protein 2 (SREBP2) and FAS gene in lipid synthesis. Otherwise, reducing the mRNA expression level of lipid synthesis genes, including SREBP1, SREBP2, fatty acid synthetase (FASN), *β*-Hydroxy *β*-methylglutaryl-CoA (HMGCR), stearoyl CoA desaturase 1 (SCD1), and increasing the mRNA expression level of lipid decomposition gene, such as carnitine palmitoyl transferease-1 (CPT1), might be involved in the lipid-lowering activity of quercetin. This study suggested that *Allium cepa* might be used to prevent and treat oxidative stress and dislipidemia-related disorders, including NAFLD.

## 1. Introduction

*Allium cepa* L.(onion), as a perennial herb of the family *Liliaceae,* is abundant and widely distributed [1,2]. Moreover, as a medicinal and edible plant, *A. cepa* is used to prevent and treat atherosclerosis, which may be developed from NAFLD [3]. *A. cepa* could effectively improve the pathological progress of NAFLD rats [4]. In addition, research has shown that *A. cepa* exhibits anti-oxidant, lipid-lowering, anti-bacterial, anti-inflammatory, and immunoprotective effects [5,6,7]. Furthermore, *A. cepa* relieves obesity and its related complications, such as hyperlipidemia, hypertension and diabetes [8]. On the other hand, as an important part of the Mediterranean diet, *A. cepa* contains not only lots of vitamins, minerals and thioamino acids, but also many biologically active constituents, such as polyphenols, phenolic acids, organic sulfur compounds, and saponins, especially quercetin and its glycosides [2,9]. Furthermore, the red variety of *A. cepa*, with the highest antioxidant and free radical scavenging activities, is a rich source of quercetin [10]. Also, quercetin reduces high blood cholesterol levels by specifically inhibiting intestinal cholesterol absorption mediated by NPC1L1 [11]. However, the experiments reported were mainly animal tests using crude extracts, and only a few studies correlated it with active constituents.

Lipid metabolism, including the synthesis, absorption, metabolism, and circulation of lipids, is regulated by a variety of enzyme proteins in the body. Among them, lipid synthesis and decomposition mainly occur in the liver [12,13]. NAFLD is a liver cell fat accumulation and fat metabolism disorder not caused by alcoholism, and is often accompanied by complications such as insulin resistance and hyperlipidemia [14], and the progress of NAFLD is closely correlated with oxidative stress and inflammatory reaction in the liver [10]. Presently, NAFLD has become a new challenge in the field of contemporary medicine with the increase in obese people and patients with type 2 diabetes, dyslipidemia and related metabolic syndromes. Nowadays, in view of the pathogenesis of NAFLD, statins and fibrates (lipid-lowering drugs), pioglitazone (insulin sensitizer), and vitamin E (antioxidant) have been used in clinical trials, but these drugs sometimes showed poor effects [15,16]. Therefore, it is of great significance to explore safe and effective strategies for the prevention and treatment of NAFLD. A large number of epidemiological studies have shown that a Mediterranean diet and consuming more fruits and vegetables, which provide natural active substances beneficial for health, such as vitamins, polysaccharides and flavonoids, are beneficial for the prevention of diseases related to NAFLD [17]. Consequently, fewer people are getting NAFLD and increasing numbers of people are being treated for NAFLD by regulating lipid metabolism using safe and effective drugs from naturally active products.

Oxidative stress, the disorder of redox balance in the body caused by excessive free radicals, might lead to a variety of diseases, including NAFLD, hyperlipidemia, diabetes, cancer, and cardiovascular and other diseases [18]. Usually, the diseases caused by excessive free radicals can be improved effectively by an appropriate amount of exogenous antioxidants or drugs such as endogenous free radical scavengers. At present, the common synthetic antioxidants are dibutyl hydroxytoluene (BHT) and tert butyl hydroquinone (TBHQ), but their toxicity to humans has resulted in people being more inclined to pay attention to natural, low-toxicity, cheap, and efficient antioxidant-active ingredients from plants. Natural antioxidant substances, such as polyphenols, vitamins, polysaccharides, and other compounds, are widely distributed and diversified, the common characteristics of which are that they contain several phenolic hydroxyl groups [19].

In this study, bioassays, including antioxidant and lipid-lowering activity in vitro, were used to guide the isolation and purification to obtain the active fractions and effective components. Furthermore, the mechanism of the lipid-lowering activity in vitro of the active fraction and the effective compound were preliminarily uncovered.

## 2. Materials and Methods

### 2.1. Materials

Fresh purple *A. cepa* was purchased from the local market in Nanchong. Ethanol of 95% purity was provided by Chengdu Shudu Industrial Co., Ltd. (Chengdu, China). Purified water was prepared by our laboratory. DPPH, K_2_S_2_O_4_, and ABTS were purchased from Aladdin. Trypsin and PBS-buffered solution were purchased from Hyclone. d-BSA, PA, and RNase remover were provided by Beijing Solabao Technology Co., Ltd. (Beijing, China). Oleic acid (OA) was purchased from Sigma–Aldrich Trading Co. Ltd. (Shanghai, China). Bicinchoninic acid (BCA) protein kits were provided by Nanjing Jiancheng Bioengineering Research Institute (Nanjing, China). Cell Counting Kit-8 (CCK-8), RIPA pyrolysis fluid, and DEPC water were purchased from Beyotime Biotechnology Co., Ltd. (Shanghai, China). Other reagents were analytical grade.

### 2.2. HepG2 Cells

HepG2 cells were provided by the Institute of Medical Imaging, North Sichuan Medical College.

### 2.3. Apparatus

An AG-9620A precision blast drying oven was produced by Shanghai Baixin Instrument and Equipment Factory. An HH-8 constant temperature water bath was purchased from Shanghai Precision Instrument Co., Ltd. (Shanghai, China). A KQ3200DB numerical control ultrasonic cleaner was provided by Kunshan Ultrasonic Instrument Co., Ltd. (Kunshan City, China). An ultrapure water machine was purchased from Sichuan YOUPU Technology Co., Ltd. (Chengdu, China). AC211S electronic analytical balance was produced by Beijing Saidoris System Co., Ltd. (Beijing, China). An electronic balance was provided by the Germany Sartorius Company. An SHZ-Ⅲ, SHB-3D circulating vacuum pump was produced by Zhengzhou Dufu Instrument Factory. A 5L rotary evaporator was purchased from Gongyi Ruide Equipment Co., Ltd. (Gongyi, China). An RE-52AA rotary evaporator was provided by Shanghai Yarong Biochemical Instrument Factory. A MULTISKAN GO full wavelength microplate reader was purchased from Thermo Fisher Scientific Company (Waltham, MA, USA). A TS100/TS100-F ECLIPSE inverted microscope was provided by Nikon Instruments Shanghai Co., Ltd. (Shanghai, China). A −80 °C ultra-low-temperature freezer was produced by American Thermo 902. An ultrasonic cell pulverizer was purchased from Ningbo Xinzhi Biotechnology Co., Ltd. (Ningbo, China). A high-speed freezing centrifuge was produced by American Thermo Mcro17R. A real-time PCR instrument and an S1000TM PCR instrument were provided by Bio-Rad.

### 2.4. Extraction and Bioassay-Guided Fractionation

The process of extraction and isolation is shown in Figure 1 and Appendix A. Firstly, 2.4 kg of dried powder of purple *A. cepa* was extracted with 64.5% of ethanol for 30 min at the temperature of 54 °C, the solid-liquid ratio being 1:10. Then, the extraction process was repeated 3 times. All the extraction solution was filtered and evaporated to obtain the extract, which weighed 1.4 kg. Subsequently, D101 Macroporous adsorption resin column chromatography(70 cm × 10 cm, 5500 mL of column volume) was used to preliminarily purify the extract of *A. cepa*. An amount of 1.38 kg of the extract was added with 1.2 L of purified water to obtain 2.1 L of sample solution. After the solution was put into the column, 22 L of purified water, 30% ethanol, 50% ethanol, 70% ethanol, and 95% ethanol were eluted, respectively. The eluted solutions were condensed to obtain the extract, which was preserved at 4 °C until content determination, activity evaluation, UPLC-MS analysis, and purification.

### 2.5. The Total Polyphenol Content and Total Flavonoid Content Determination of the Extract and Purified Fractions from A. cepa

#### 2.5.1. Preparation of the Standard Solution

A total of 10 mg of gallic acid (Beijing Solabao Technology Co., Ltd.) was precisely weighed, and completely dissolved in purified water. After the solution was cooled, it was maintained at a constant volume of 100 mL within a volumetric flask to obtain 0.1 mg/mL solution. An amount of 10 mg of rutin (Beijing Solabao Technology Co., Ltd.) was precisely weighed, and completely dissolved in 70% ethanol. Then, the solution was maintained at a constant volume of 100 mL within a volumetric flask to obtain 0.1 mg/mL solution. The two solutions were both kept at 4 °C before use.

#### 2.5.2. Preparation of the Sample Solution

Amounts of 0.1500 g of ACE, ACA, AC30%, AC50%, AC70%, and AC95% were precisely weighed, respectively, and completely dissolved in purified water. Then, the solutions were maintained at a constant volume of 100 mL within a volumetric flask to obtain 1.5 mg/mL solution, respectively. After the solutions were filtered using a filter membrane of 0.45 μm, they were kept at 4 °C until use.

#### 2.5.3. Preparation of the Folin Reagent and Other Reagents Related to Content Determination

The Folin reagent was prepared according to the literature [20]. The other reagents related to the content determination were also prepared following the previous practices in the literature [20,21].

#### 2.5.4. The Total Polyphenol Content Determination of the Extract and Purified Fractions from *A. cepa*

The total polyphenol content determination of the extract and purified fractions from *A. cepa* was analyzed based on the literature [20]. The standard curve was established using gallic acid as the standard, and the regression equation was Y = 161.92x − 0.0275 (R^2^ = 0.993) with the linear range of (0.001–0.005) mg/mL.

#### 2.5.5. The Total Flavonoid Content Determination of the Extract and Purified Fractions from *A. cepa*

The total flavonoid content determination of the extract and purified fractions from *A. cepa* was investigated according to the literature [21]. The standard curve was established using rutin as the standard, and the regression equation was Y = 25.177x + 0.0124 (R^2^ = 0.9936) with the linear range of (0.008–0.040) mg/mL.

### 2.6. The Antioxidant Activity Evaluation

#### 2.6.1. Preparation of the Standard and Sample Solutions

Amounts of 4.1 mg of rutin, 74.4 mg of ACE, 50.3 mg of ACA, 35 mg of AC30%, 97.8 mg of AC50%, 15.5 mg of AC70%, and 10 mg of AC95% were precisely weighed and dissolved in DMSO to obtain solutions of 10 mg/mL. All the solutions were kept at 4 °C before use.

#### 2.6.2. Preparation of ABTS^+●^ and DPPH^●^ Solutions

The preparation of ABTS^+●^ and DPPH^●^ solutions was carried out according to the literature [22].

#### 2.6.3. Evaluation of ABTS^+●^ and DPPH^●^ Free Radicals Scavenging Activity

The evaluation of ABTS^+●^ and DPPH^●^ free radicals scavenging activity was conducted using a method based on the literature [22].

The details of the groups were as follows: (1)Negative group: the concentrations of DMSO and ethanol were the same with those of the experimental groups. Furthermore, the OD values were controlled at the range of 0.70 ± 0.02.(2)Experimental groups: the final concentrations of ACE, AC30%, AC50%, AC70%, and AC95% were 100, 50, and 25 μg/mL, of which the diluted solvent was ethanol. When the inhibition rate (100 μg/mL) was more than 50%, five final concentrations (double dilution) of this extract or fractions were carried out to calculate the values of IC_50_.(3)Positive group: the final concentrations of rutin were 100, 50, and 25 μg/mL, of which the diluted solvent was ethanol.

### 2.7. The Lipid-Lowering Activity Evaluation

#### 2.7.1. Preparation of the Solutions

ACE, ACA, AC30%, AC50%, AC70%, and AC95% were precisely weighed, and 200 μL of DMSO plus 3.8 mL of purified water were added to obtain the solutions, i.e., 22.125 mg/mL of ACE, 32.95 mg/mL of ACA, 15.225 mg/mL of AC30%, 13.975 mg/mL of AC50%, 3.225 mg/mL of AC70%, and 5.075 mg/mL of AC95%. The preparation of simvastatin was completed on the basis of the literature [23].

#### 2.7.2. Cell Culture

The complete medium was composed of 89% of high-glucose DMEM medium (Hyclone), 10% of heat-killed FBS (Gibco, Grand Island, NY, USA), and 1% of Penicillin-streptomycin (Hyclone, Logan, UT, USA). HepG2 cells were cultured in the incubator (3111CO_2_, Thermo Forma company, Marietta, OH, USA) at 5% CO_2_, 37 °C, with saturated humidity. 

#### 2.7.3. Experimental Groups of Cell Viability Assay

The experimental groups were diluted using culture medium of 3 final concentrations including high, medium, and low concentrations, i.e., ACE (86.29, 172.58, 258.87 μg/mL), ACA (76.86, 153.72, 230.58 μg/mL), AC30% (5.07, 10.14, 15.22 μg/mL), AC50% (3.99, 7.98, 11.98 μg/mL), AC70% (0.44, 0.88, 1.31 μg/mL), and AC95% (1.00, 2.00, 3.00 μg/mL), all of which possessed the same raw drug concentrations. 

#### 2.7.4. Cell Viability Assay

The cell viability experiment was conducted according to the literature [23]. Briefly, 100 μL of the cells growing in logarithmic phase were inoculated in 96-well plates. After 24 h, the medium was discarded. Then, the high-fat models were induced using 1 mM of FFA treated for 24 h. On the basis of the experimental groups of Section 2.7.3, each well of the 96-well plates was added with 100 μL of culture medium with or without test drugs. After 24 h, 110 μL of CCK-8 solution, with 10:1 ratio of basic culture medium and CCK-8, was added. After the cells were incubated for 1.5 h at the temperature of 37 °C, the OD values were measured at 450 nm using MULTISKAN GO full wavelength microplate reader. All the experiments were repeated in triplicate.

#### 2.7.5. Preparation of Free Fatty Acid

The free fatty acid (FFA) was prepared according to the literature [23,24].

#### 2.7.6. Induction of In Vitro Hepatic Steatosis Models

The induction of in vitro hepatic steatosis models was carried out according to the literature [23].

#### 2.7.7. Experimental Groups of Lipid-Lowering Effect

The experimental groups correlated with *A. cepa* were the same with Section 2.7.3. except for the negative group and positive group as follows. 

(1)Negative group: the blank control was diluted, using culture medium, to a final solution (1% d-BSA).(2)Positive group: the final concentration of simvastatin was 2 μg/mL (1.0 mM FFA).

#### 2.7.8. Oil Red O Staining

The oil red O staining was obtained using a method based on the literature [23].

#### 2.7.9. Evaluation of Intracellular TC and TG Content

An amount of 2 mL of the cells (about 5 × 10^5^ cells/well) growing in logarithmic phase were inoculated in 6-well plates. The culture medium was discarded when the density of the cells reached 60–70% (generally 24 h later). After the cells were washed 2 times using PBS solution, according to the groups in Section 2.7.7, into each well of the 6-well plates 2 mL of culture medium was added with 1 mM of FFA induced for 24 h to establish the high-fat models, along with or without test drugs. Then, oil red O staining was performed according to the kit instructions (Nanjing Jiancheng Bioengineering Research Institute, Nanjing, China). All the experiments were repeated 3 times. 

#### 2.7.10. Detection of the Gene Expression Related to Lipids of the Lipid-Lowering Fraction and Compound Using RT-qPCR

According to the groups in Section 2.7.7, the concentrations of AC30% and quercetin were 5.07 μg/mL and 12.5 μg/mL, respectively. After 24 h, the total RNA was extracted using a traditional Trizol test kit (Beijing Solabao Technology Co., Ltd.). Then, the cDNA was synthesized according to the reverse transcription Kit (Saiguo Biotechnology Co., Ltd., Guangzhou, China) instructions. Subsequently, PCR amplification was performed using the primers described in Table 1. Firstly, 10 μL reaction system was prepared according to the kit Bestar^®^ SybrGreen qPCR Mastermix (Shanghai Xinghan Biotechnology Co., Ltd., Shanghai, China). The RT-qPCR process was as follows. Firstly, the reaction system was predenaturated for 3 min at 95 °C, and then denaturated for 10 s at 95 °C. Subsequently, the reaction system was annealed for 30 s at 60 °C, and then elongated for 30 s at 72 °C. This process was repeated 39 times. Dissolution curve defaulted by the instrument, i.e., with the gradient of 0.5 °C from 65 °C to 95 °C, was used. Relative expression levels equaled 2^(−ΔΔCt)^, of which ΔΔCt equaled (Ct_Target gene_ − Ct_Internal gene_)_experiment group_ − (Ct_Target gene_ − Ct _Internal gene_)_control group_ (Ct represented the number of cycles in which the target amplification product reached the set threshold). 

### 2.8. UPLC-MS Analysis of Bioactive AC30% and AC50% Fractions

#### 2.8.1. Sample Preparation

The fractions AC30% and AC50% were dissolved in chromatographic grade methanol, and then filtered using a filter membrane of 0.45 μm(organic system, Tianjin Jinteng Experimental Equipment Co., Ltd., Tianjin, China) to obtain the sample solutions preserved at 4 °C before use.

#### 2.8.2. Liquid Chromatography and Mass Spectrum Conditions

Agilent 1290 Ultra High Performance Liquid Chromatography (UPLC) equipped with Chromatographic column and waters BEH C18 (2.1 × 100 mm, 1.7 μm) were used. Mobile phase A, 0.1% acetic acid aqueous solution; Mobile phase B, acetonitrile; flow rate, 0.3 mL/min; injection volume, 2 μL; column temperature, 35 °C; detection wavelength, UV 254 nm; gradient elution conditions were shown in Table 2. Meanwhile, an Agilent mass spectrum instrument (Agilent Technology Co., Ltd., Santa Clara, CA, USA) was utilized, with the parameters of electrospray ionization source (ESI), negative ion scanning modewith −3200 V of voltage, 100–1500 *m*/*z* of one-level mass scanning scope, 350 °C of sheath gas temperature, and 12 L/min of sheath gas flow.

### 2.9. Separation of Bioactive Fractions 

#### 2.9.1. Purification of the Compounds Using Sephadex LH-20 and Preparative HPLC

Sephadex LH-20 (LH-20, GE Healthcare Bio-Sciences, Uppsala, Sweden) column chromatography was used to separate AC30% fraction. A total of 77.3 g of AC30% fraction was dissolved using methanol and filtered to obtain a sample solution of 250 mL. After the sample solution was added to the column, methanol was used as the elution solvent to obtain 235 fractions using penicillin bottles. Then, thin layer chromatography was used to detect the separation situation and further merge them together. Finally, preparative high performance liquid chromatography (PHPLC) was used to purify the fractions to obtain the compounds. PHPLC chromatography conditions were as follows: constant gradient of 5% or 30%, etc., chromatography column of 30 × 250 mm, detection wavelength of 210 nm and 254 nm, flow speed of 20 or 10 mL/min, and injection volume of 1000–3000 μL. 

#### 2.9.2. Purity Analysis of the Compounds

The compounds, obtained through D101 macroporous adsorption resin column chromatography, Sephadex LH-20 column chromatography, and PHPLC, were added to HPLC using respective injection (only compounds) and mixed injection (compound and the fraction together), the aim of which was to judge the retention time of the compound in HPLC and analyze the purification degree of the compound. The chromatography conditions were as follows: 0–70 min, 5–100% methanol; 70–100 min, 100% methanol, chromatography column of 30 × 250 mm, detection wavelength of 210 nm and 254 nm, flow speed of 20 or 10 mL/min, and injection volume of 500 μL.

#### 2.9.3. MS Identification

Compounds **1**–**5** were dissolved in methanol to obtain the molecular weights using MS (LC-MS, Shimadzu Enterprise China Co., Ltd., Suzhou, China).

#### 2.9.4. NMR Identification

Compounds **1**–**5** were dissolved in DMSO-d_6_ to obtain the structures using NMR (Nuclear Magnetic Resonance instrument, Bruker company Fällanden, Switzerland). 

### 2.10. Biological Evaluation of the Purified Constituents

#### 2.10.1. Antioxidant Activity Evaluation of the Isolated Components

The method is the same as that in Section 2.6 except for the preparation of the sample solutions. Briefly, Compounds **1**–**5** (Compound **1**, gallic acid; Compound **2**, quercetin-3,4’-diglucoside; Compound **3**, Isoquercetin; Compound **4**, quercetin; Compound **5**, Isorhamnetin) were dissolved in DMSO to obtain 50 mg/mL of solutions. All the solutions were kept at 4 °C until use.

#### 2.10.2. Lipid-Lowering Activity Investigation of the Purified Compounds

The method is the same with that in Section 2.7 except for the preparation of the sample solutions and the experimental groups. As far as the preparation of the sample solutions was concerned, Compounds **1**–**5** (Compound **1**, gallic acid; Compound **2**, quercetin-3,4-diglucoside; Compound **3**, Isoquercetin; Compound **4**, quercetin; Compound **5**, Isorhamnetin) were dissolved in DMSO to obtain 50 mg/mL of solutions. After all the solutions were filtered using a filter membrane (Millipak) of 0.22 μm, they were kept at −20 °C until use. Furthermore, for the experimental groups, Compounds **1**–**5** (Compound **1**, gallic acid; Compound **2**, quercetin-3,4-diglucoside; Compound **3**, Isoquercetin; Compound **4**, quercetin; Compound **5**, Isorhamnetin) were dissolved using the culture medium to obtain 50, 25, and 12.5 μg/mL of solutions (1.0 mM FFA). 

### 2.11. Statistical Analysis

Office Excel 2019 was used for the primitive data dealing. MestReNova 14.2.0 was utilized to deal with the data related to UPLC-MS, ^1^H-NMR, and ^13^C-NMR. Chemical structures were drawn with the use of ChemDraw 19.0, and adobe Photoshop CS4 was used to merge the images with high resolution. All the experiments related to the activity were repeated 3 times, and the results were expressed as $\bar{x}$ ± S. GraphPad Prism 8.3.0 was used to analyze the data correlated with the antioxidant and lipid-lowering action, including ANOVA, Brown–Forsythe, and Welch ANOVA test. ANOVA was used for the comparison of the data between groups. Brown–Forsythe and Welch ANOVA tests were used to determine the significance of the differences between the average values. *p* < 0.05 meant that it was statistically significant. 

## 3. Results and Discussion

### 3.1. Bioassay-Guided Isolation and Characterization of the Most Active Fractions from A. cepa

In folk medicine, *Allium cepa* is used to treat atherosclerosis [3]. Also, hyperlipidemia is one of the most high risk factors of atherosclerosis [25]. Furthermore, NAFLD is often accompanied with hyperlipidemia [14]. In addition, the progress of NAFLD is closely correlated with oxidative stress [26]. In a previous study, it has been shown that *A. cepa* exhibits lipid-lowering action in vivo [27]. Therefore, in this study, bioassays, including antioxidant effect and lipid-lowering activity in vitro, were used to guide the isolation and purification to obtain the active fractions and constituents. After solvent extraction, D101 macroporous adsorption resin column chromatography was used to preliminarily purify the extract to obtain five fractions, i.e., ACA, AC30%, AC50%, AC70%, and AC95%.

### 3.2. Determination TPC and TFC of the Extract and Fractions from Allium cepa

The literature has shown that *A. cepa* is abundant in polyphenols [28]. Meanwhile, polyphenols mainly include flavonoids, phenolic acids, and coumarins [29]. One of the most striking characteristics of polyphenols is their antioxidant activity. Oxidative stress was believed to play an important role in the pathogenesis of NAFLD [30]. Therefore, this study investigated the TPC and TFC of *A.cepa*. From Table 3, we can see that, followed by AC50%, AC30% possessed a higher total polyphenol content, except for the ACA fraction, which might contain proteins and tannins. Furthermore, AC30% exhibited the highest total flavonoid content, followed by AC50%. The previous study showed that the TPC of *A. cepa* extract varied from 4.6 to 74.1 mg/g GAE [10]. Furthermore, this experiment uncovered the TPC and TFC of the different fractions from *A. cepa*.

### 3.3. Antioxidant Activity of the Extract and Fractions from Allium cepa

In this section, ABTS^+●^ and DPPH^●^ free radicals were used to evaluate the antioxidant activity. First of all, three concentrations of 100, 50, and 25 μg/mL were used to scavenge the two kinds of free radicals. From the results of Figure 2 and Figure 3, we can see that AC30%, AC50%, and AC70% exhibited better antioxidant activity. Thus, the three fractions were used to measure the IC_50_ values. As shown in Table 4, AC50% possessed the best free scavenging activity. Correspondingly, AC50% was preliminarily taken as the active fraction with antioxidant activity from *A. cepa*. However, another study reported the antioxidant fractions of *A. cepa* through flash chromatography, instead of D101 macroporous adsorption resin column chromatography [31]. 

### 3.4. Lipid-Lowering Action and Its Mechanism of the Extract and Fractions from Allium cepa

#### 3.4.1. Cell Viability 

As is knowm, the lipid-lowering activity in vitro can be evaluated by the reduction in the levels of TC and TG. However, the drop of the levels of TG and TC in vitro could also be also caused by the death of the cells. Therefore, a CCK-8 test was used to evaluate the cell viability. As shown in Figure 4, through previous pretest, the cell viability rates of all the experimental groups were more than 90% at different concentrations, including high, medium, and low concentrations of the extract and fractions. Therefore, all of the concentrations of the experiment groups were used to evaluate the following lipid-lowering activity.

#### 3.4.2. Establishment of In Vitro Hepatic Steatosis Models

In this section, HepG2 cells were treated by FFA for 24 h to establish the in vitro hepatic steatosis models according to the literature [23,24]. The fats in the model group exhibited red or orange, which was more obvious than the negative group. Therefore, the in vitro hepatic steatosis models were successfully established.

#### 3.4.3. Evaluation of the Lipid-Lowering of Extract and Fractions against In Vitro Hepatic Steatosis Models Using Oil Red O Staining 

As shown in Figure 5, the fats exhibited red or orange. Furthermore, the fat droplets in the negative and model groups were more obvious than those in the experimental and positive groups.

#### 3.4.4. Measurement of TC and TG Levels of the Extract and Fractions against In Vitro Hepatic Steatosis Models

Before measuring the levels of TC and TG, the standard curves of TC, TG, and total protein were established. As far as TC was concerned, the standard curve was y = 0.0029x + 0.0563 (R^2^ = 0.9905) with the linear range of 0–51.7 mmoL/L. As for TG, the standard curve was y = 0.0129x + 0.0941 (R^2^= 0.9932) with the linear range of 0–22.6 mmoL/L. For total protein, the standard curve was y = 0.0001x + 0.1608 (R^2^ = 0.9978) with the linear range of 0–5.24 g/L. As shown in Figure 6 and Figure 7, the intracellular levels of TG and TC in model group increased greatly, with *p* < 0.001 and *p* < 0.05, respectively. The levels of TG and TC of all the experimental groups decreased to some extent, among which the AC30% group exhibited the best effect in reducing the TG and TC levels, with *p* < 0.001 and *p* < 0.05, respectively. Correspondingly, AC30% was the active fraction with lipid-lowering activity, the mechanism of which was subsequently investigated. The previous study showed that *A. cepa* extract reduced the levels of TC and TG in vivo [27]. This experiment further discovered that AC30% was the lipid-lowering fraction of *A. cepa*.

#### 3.4.5. Effect of AC30% on 10 Genes Related to Lipids

The balance of lipids synthesis and decomposition was a process with complex regulations. With the absorption of the energy, AMPK was activated, and then SREBPs, PPARs, LXRs, and ACC in the way of lipids metabolism were initiated. Furthermore, SREBPs and ACC were inhibited, and PPARs as well as LXRs were activated. Subsequently, the levels of FAS, SCD2, and HMGCR followed by SREBPs decreased, and then the activity of both CPT1 followed by PPARs, CYP7A1, and LXRs increased [32]. Conversely, the activation and inhibition of the followed target genes could regulate the above genes, such as SREBPs, PPARs, LXRs, and ACC [33]. Thus, these genes related to lipids synthesis and disintegration, such as SREBP1, SREBP2, ACC1, FASN, HMGCR, SCD1, CYP7A1, PPARα, LXRα, and CPT1 were used to investigate the mechanism with the lipid-lowering effect of AC30% fraction. The results of RT-qPCR showed that the parallelism of amplification curves correlated with these 10 genes were perfect, and the amplification products were sole and clear. As shown in Figure 8, compared with the negative group, the synthesis genes of the model group increased and the disintegration genes were activated. Compared with the model group, the mRNA levels of the synthesis genes in the group of the AC30% fraction, such as SREBP1, SREBP2, ACC1, FASN, HMGCR, and SCD1, decreased to some extent, especially SREBP2 (*p* < 0.05) and FAS (*p* < 0.01).

### 3.5. Analysis of AC30% and AC50% from Allium cepa Using UPLC-MS

#### 3.5.1. UPLC-MS Analysis of AC 30%

In Section 3.4.4, we saw that AC30% was the active fraction with a lipid-lowering effect. Then, UPLC-MS was used to preliminarily analyze the chemical constituents and establish the fingerprint chromatogram. AC30% showed a perfect response under the negative ion model (Figure 9 and Figure 10). From the UPLC-ESI-MS scanning results (Figure 9 and Figure 10), we could obtain the structural information of some components of AC30%. By comparing the UPLC-UV chromatogram and ESI-MS mass spectrum, and referring to the related literature, we preliminarily speculated eight molecules, including six polyphenols, mainly within 20–50 min of the total ion flow chromatogram. Furthermore, as shown in Table 5, Figure 9, Figure 10 and Figure 11 and Appendix A, based on [M-H]^-^ and MS/MS data from ESI-MS of these peaks, the main compounds of AC30% were 3,4-dihydroxybenzoic acid (20.02 min) and Quercetin-3,4′-*O*-diglucoside (48.58 min) [34,35]. This research was the same as a previous study, which showed that the antioxidant fraction of *A. cepa*, through flash chromatography, was also abundant in protocatechuic acid and quercetin-3,4′-O-diglucoside. 

#### 3.5.2. UPLC-MS Analysis of AC50%

Similarly, in Section 3.3, we saw that AC50% was the active fraction with antioxidant activity. Then, UPLC-MS was used to preliminarily analyze the chemical constituents and establish the fingerprint chromatogram of AC50%. It showed an ideal response under the negative ion model (Figure 12 and Figure 13). From the UPLC-ESI-MS scanning results (Figure 12 and Figure 13), we obtained the structural information of some components of AC50%. By comparing the UPLC-UV chromatogram and ESI-MS mass spectrum, and referring to the related literature, we preliminarily speculated eight constituents, including six polyphenols, within 35–60 min of the total ion flow chromatogram. Moreover, as shown in Table 6, Figure 12, Figure 13 and Figure 14 and Appendix A, according to [M-H]^−^ and MS/MS data from ESI-MS of these peaks, the main compounds of AC50% were Quercetin-3-*O*-glucoside (52.38 min) and Quercetin (58.04 min) [35]. This result was the same as a previous study, which showed that *A. cepa* extract, abundant in quercetin and isoquercitrin, exhibited in vivo lipid-lowering action. 

### 3.6. Isolation and Elucidation of the Compounds from Allium cepa

Except for the UPLC-MS analysis of AC30% and AC50%, bio-assay-guided isolation was carried out to obtain the active compounds with the lipid-lowering and antioxidant activities. As a result, as shown in Figure 15, five constituents were obtained. The MS and NMR data were exhibited as follows, with a purity of more than 98%.

Compound **1**, white powder, ESI-MS *m*/*z*: 153 [M-H]^-^;^1^H-NMR (600 MHz, DMSO-d_6_), δ: 7.33(d, *J* = 1.8 Hz, 1H, H-2), 7.28(dd, *J* = 1.8,7.8 Hz, 1H, H-6), 6.77 (d, *J* = 7.8 Hz,1H, H-5). The data were the same in the literature [37,38]. Therefore, compound **1** was identified as gallic acid. The spectra are shown in Appendix A.

Compound **2**, yellow powder, ESI-MS *m*/*z*: 625[M-H]^−^; ^1^H-NMR (400 MHz, DMSO-d_6_), δ: 7.65(d, *J* = 1.6 Hz, 1H, H-2′), 7.61(dd, *J* = 2.0,8.8 Hz, 1H, H-6′), 7.20(d, H = 8.8 Hz, 1H, H-5′), 6.45(d, *J* = 1.6 Hz, 1H, H-8), 6.22(d, *J* = 2.4 Hz, 1H, H-6), 5.50(d, *J* = 7.6 Hz, 1H, H-1″), 4.87(d, *J* = 7.6 Hz, 1H,H-1‴), 3.03–3.76(m, H-2″~6″,2‴~6‴). The data were the same in the literature [38,39]. Therefore, compound **2** was identified as quercetin-3,4-*O*-di glucoside. The spectra are shown in Appendix A.

Compound **3**, yellow powder, ESI-MS *m*/*z*: 463[M-H]^−^; Separate injection (only compound 3) and mixed injection (compound 3 and the standard isoquercetin) into HPLC showed that compound 3 was pure and identified as isoquercetin. The spectra are shown in Appendix A. 

Compound **4**, yellow powder, ESI-MS *m*/*z*: 301[M-H]^-^;^1^H-NMR (400 MHz, DMSO-d_6_), δ: 7.67(brs,1H, H-2′), 7.54(dd, *J* = 1.6, 8.4 Hz, 1H, H-6′), 6.88(d, *J* = 8.8 Hz, 1H.H-5′), 6.40(brs, 1H, H-8), 6.18(brs, 1H, H-6). The data were the same in the literature [38,39]. Therefore, compound **4** was identified as quercetin. The spectra are shown in Appendix A.

Compound **5**, yellow powder, ESI-MS *m*/*z*: 315[M-H]^−^; 1H NMR (400 MHz, DMSO-d_6_) δ: 7.76(brs, 1H, H-2′), 7.69(d, *J* = 7.2 Hz, 1H, H-6′), 6.94(d, *J* = 8.4 Hz, 1H, H-5′), 6.48(brs, 1H, H-8), 6.20(brs, 1H, H-6), 3.84(s, 3H, –OCH_3_). The data were the same in the literature [38,39]. Thus, compound **5** was identified as isorhamnetin. The spectra are shown in Appendix A.

### 3.7. Antioxidant Activity of the Compounds from A. cepa 

As shown in Figure 16, the inhibition rate of compounds **1**, **3**, and **4** were more than 50%. In Figure 17, however, the inhibition rates of compounds **1**–**5** were more than 50%. Therefore, the IC_50_ values of the five compounds were further measured. As shown in Table 7, compound 4, quercetin, exhibited the best antioxidant activity with an IC_50_ value of 7.12 μg/mL and 2.121 μg/mL against DPPH^●^ and ABTS^+●^ free radicals, respectively. It was considered that the antioxidant potency of these phenolic antioxidants was proportional to the number of -OH groups present in the aromatic ring(s) [40]. However, how to exhibit the antioxidant action of these -OH groups still needs to be further comprehensively studied.

### 3.8. Lipid-Lowering Action and Its Mechanism of the Compounds from Allium cepa

#### 3.8.1. Cell Viability

As shown in Figure 18, the cell viability rates of all the experimental groups were over 90% at different concentrations, including high, medium, and low concentrations, of the compounds. Therefore, all of the concentrations of the constituents were applied to the lipid-lowering activity evaluation as follows. 

#### 3.8.2. Measurement of TC and TG Levels of Compounds **1**–**5** against In Vitro Hepatic Steatosis Models

The standard curves of TG and TC were the same with Section 3.4.4. As shown in Figure 19 and Figure 20, the intracellular levels of TG and TC in the model group increased greatly, with *p* < 0.001 and *p* < 0.05, respectively. The levels of TG and TC of all the five polyphenols decreased to some extent, among which quercetin exhibited the best effect with decreases in TG (*p* < 0.01) and TC (*p* < 0.0001), respectively. Therefore, quercetin was the active moner with lipid-lowering activity, the mechanism of which was subsequently investigated. 

#### 3.8.3. Effect of Quercetin on 10 Genes Related to Lipids

As shown in Section 3.8.1 and Section 3.8.2, the lipid-lowering constituent of *A. cepa* was quercetin. Correspondingly, an RT-qPCR experiment was performed to uncover the lipid-lowering mechanism of quercetin. The RT-qPCR results of quercetin are shown in Figure 21. Some of the mRNA levels correlated with the lipids synthesis obviously decreased, such as SREBP1 (*p* < 0.05), SREBP2 (*p* < 0.0001), FAS (*p* < 0.01), HMGCR (*p* < 0.01), and SCD1 (*p* < 0.05). Some of the mRNA levels related to the lipids decomposition obviously increased, including CPT1 (*p* < 0.0001). As shown in Section 3.7, the antioxidant substance was also quercetin. Due to various reasons, we only isolated five major polyphenols, which were further used to evaluate the activities. 

## 4. Conclusions

In conclusion, the active fractions and constituents with the antioxidant and lipid-lowering activity from *A. cepa* were investigated using bioassay-guided isolation and identification, which helped to uncover the substantial basis of the prevention and treatment of atherosclerosis in folk medicine. The antioxidant fraction of *A. cepa* was AC50%, the main constituents of which were quercetin and isoquercitrin, as discovered by way of UPLC-MS analysis, the incorporation of an investigation of the literature regarding the compounds related to *A. cepa*, and constituents characterization. Similarly, the lipid-lowering active fraction of *A. cepa* was AC30% with the main constituent of 3,4-dihydroxybenzoic acid and quercetin 3,4′-*O*-diglucoside. Furthermore, antioxidant and lipid-lowering-activity-guided isolation led to the isolation and identification of five known compounds, and quercetin was both the best free radical scavenger and lipid-lowering constituent. Moreover, the mechanism of the lipid lowering of AC30% might be related to its reduction in mRNA expression levels of the SREBP2 and FAS genes in lipid synthesis. Otherwise, the lipid-lowering activity of quercetin might be involved in reducing the mRNA expression level of the lipid synthesis gene, including SREBP1, SREBP2, FAS, HMGCR, and SCD1, and increasing the mRNA expression level of a lipid decomposition gene, such as CPT1. However, these effective fractions and constituents with antioxidant and lipid-lowering actions should be further verified by performing in vivo experiments. This study provided a scientific pharmacodynamic material basis of the use of *A. cepa* for the prevention and treatment of oxidative stress and dislipidemia-related disorders, including NAFLD.

## Figures and Tables

**Figure 1 antioxidants-12-01448-f001:**
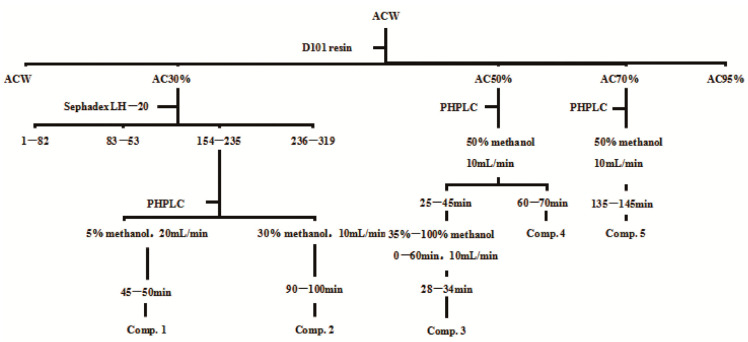
Diagram of the extraction, separation, and purification of *Allium cepa.* Note: ACE, the ethanol extract of *Allium cepa*; ACW, the water fraction of ACE through D101 macroporous adsorption resin column chromatography; AC30%, the 30% ethanol fraction of ACE through D101 macroporous adsorption resin column chromatography; AC50%, the 50% ethanol fraction of ACE through D101 macroporous adsorption resincolumn chromatography; AC70%, the 70% ethanol fraction of ACE through D101 macroporous adsorption resin column chromatography; AC95%, the 95% ethanol fraction of ACE through D101 macroporous adsorption resin column chromatography. Compound **1**, gallic acid; Compound **2**, quercetin-3,4’-diglucoside; Compound **3**, Isoquercetin; Compound **4**, quercetin; Compound **5**, Isorhamnetin.

**Figure 2 antioxidants-12-01448-f002:**
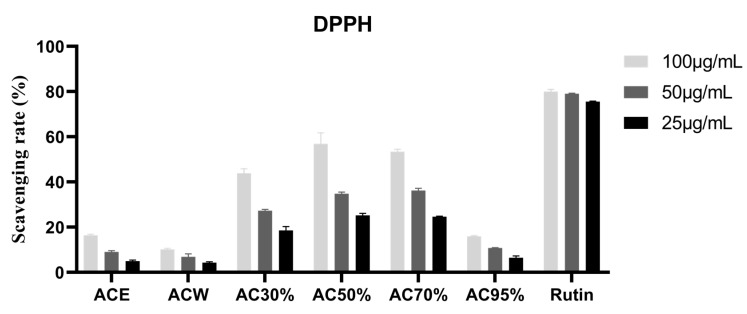
DPPH^●^ scavenging rate of the fractions from *Allium cepa*. Note: ACE, the ethanol extract of *Allium cepa*; ACW, the water fraction of ACE through D101 macroporous adsorption resin column chromatography; AC30%, the 30% ethanol fraction of ACE through D101 macroporous adsorption resin column chromatography; AC50%, the 50% ethanol fraction of ACE through D101 macroporous adsorption resin column chromatography; AC70%, the 70% ethanol fraction of ACE through D101 macroporous adsorption resin column chromatography; AC95%, the 95% ethanol fraction of ACE through D101 macroporous adsorption resin column chromatography.

**Figure 3 antioxidants-12-01448-f003:**
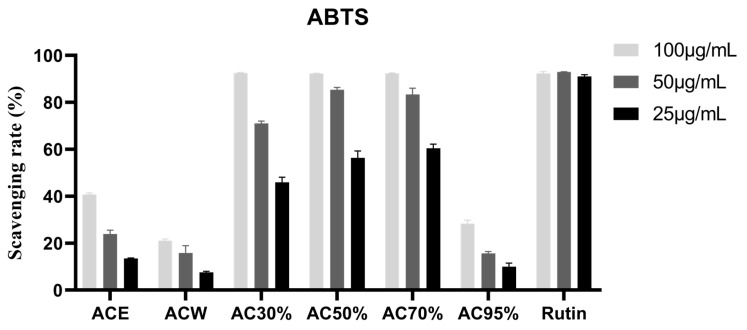
ABTS scavenging rate of the fractions from *Allium cepa*. Note: ACE, the ethanol extract of *Allium cepa*; ACW, the water fraction of ACE through D101 macroporous adsorption resin column chromatography; AC30%, the 30% ethanol fraction of ACE through D101 macroporous adsorption resin column chromatography; AC50%, the 50% ethanol fraction of ACE through D101 macroporous adsorption resin column chromatography; AC70%, the 70% ethanol fraction of ACE through D101 macroporous adsorption resin column chromatography; AC95%, the 95% ethanol fraction of ACE through D101 macroporous adsorption resin column chromatography.

**Figure 4 antioxidants-12-01448-f004:**
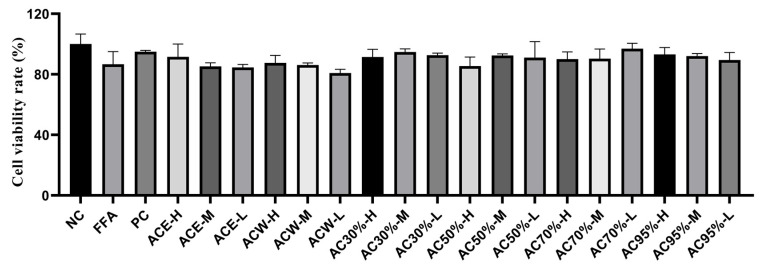
Results of cell survival test of the fractions from *Allium cepa*. Note: NC, the negative group; FFA, the model group; PC, the positive group; ACE, the ethanol extract of *Allium cepa*; ACW, the water fraction of ACE through D101 macroporous adsorption resin column chromatography; AC30%, the 30% ethanol fraction of ACE through D101 macroporous adsorption resin column chromatography; AC50%, the 50% ethanol fraction of ACE through D101 macroporous adsorption resin column chromatography; AC70%, the 70% ethanol fraction of ACE through D101 macroporous adsorption resin column chromatography; AC95%, the 95% ethanol fraction of ACE through D101 macroporous adsorption resin column chromatography.

**Figure 5 antioxidants-12-01448-f005:**
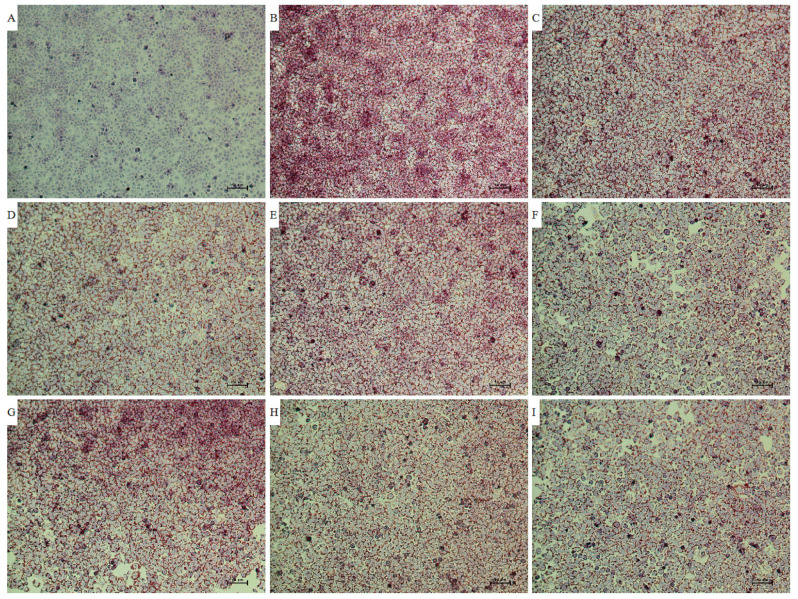
Representative pictures of oil red O staining of the extract and fractions from *Allium cepa* L. and related controls. Note: (**A**), the negative group; (**B**), the model group; (**C**), the positive group; (**D**), high-dose group of *A. cepa* ethanol extract; (**E**), high-dose group of the water fraction of ACE through D101 macroporous adsorption resin column chromatography; (**F**), high-dose group of the 30% ethanol fraction of ACE through D101 macroporous adsorption resin column chromatography; (**G**), high-dose group of the 50% ethanol fraction of ACE through D101 macroporous adsorption resin column chromatography; (**H**), high-dose group of the 70% ethanol fraction of ACE through D101 macroporous adsorption resin column chromatography; (**I**), high-dose group of the 95% ethanol fraction of ACE through D101 macroporous adsorption resincolumn chromatography.

**Figure 6 antioxidants-12-01448-f006:**
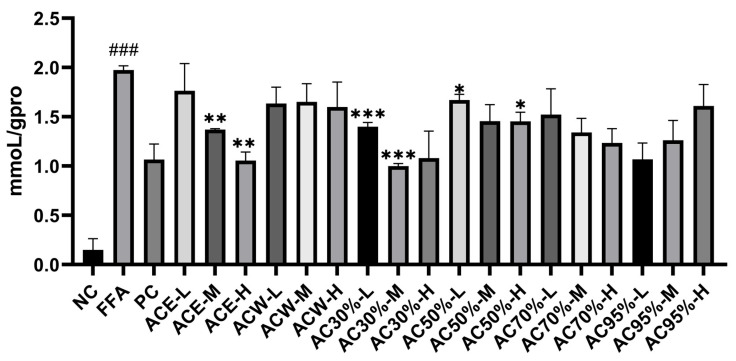
TG levels in HepG2 cells treated for 24 h with the extract and fractions from *Allium cepa*. Note: NC, the negative group; FFA, the model group; PC, the positive group; ACE, the ethanol extract of *Allium cepa*; ACW, the water fraction of ACE through D101 macroporous adsorption resin column chromatography; AC30%, the 30% ethanol fraction of ACE through D101 macroporous adsorption resin column chromatography; AC50%, the 50% ethanol fraction of ACE through D101 macroporous adsorption resincolumn chromatography; AC70%, the 70% ethanol fraction of ACE through D101 macroporous adsorption resin column chromatography; AC95%, the 95% ethanol fraction of ACE through D101 macroporous adsorption resin column chromatography. H, M, and L, respectively, indicate high, medium, and low concentrations of the experimental groups; * *p* < 0.05, ** *p* < 0.01, and *** *p* < 0.001 as compared to the model group of FFA-induced cells; ### *p* < 0.001 as compared to the negative controls.

**Figure 7 antioxidants-12-01448-f007:**
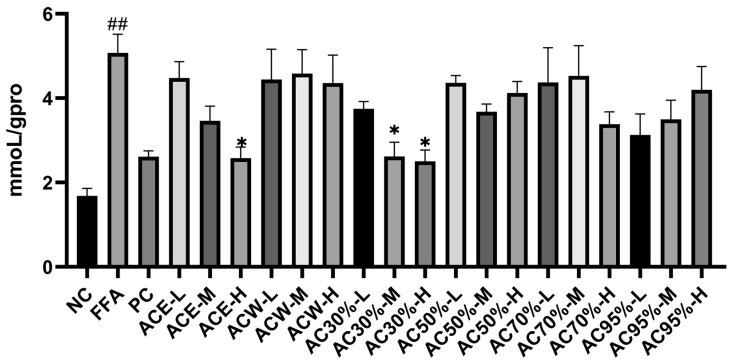
TC levels in HepG2 cells treated for 24 h with the extract and fractions from *Allium cepa*. Note: NC, the negative group; FFA, the model group; PC, the positive group; ACE, the ethanol extract of *Allium cepa*; ACW, the water fraction of ACE through D101 macroporous adsorption resin column chromatography; AC30%, the 30% ethanol fraction of ACE through D101 macroporous adsorption resin column chromatography; AC50%, the 50% ethanol fraction of ACE through D101 macroporous adsorption resin column chromatography; AC70%, the 70% ethanol fraction of ACE through D101 macroporous adsorption resin column chromatography; AC95%, the 95% ethanol fraction of ACE through D101 macroporous adsorption resin column chromatography. H, M, and L, respectively, indicate high, medium, and low concentrations of the experimental groups; * *p* < 0.05 as compared to the model group of FFA-induced cells; ## *p* < 0.01 as compared to the negative controls.

**Figure 8 antioxidants-12-01448-f008:**
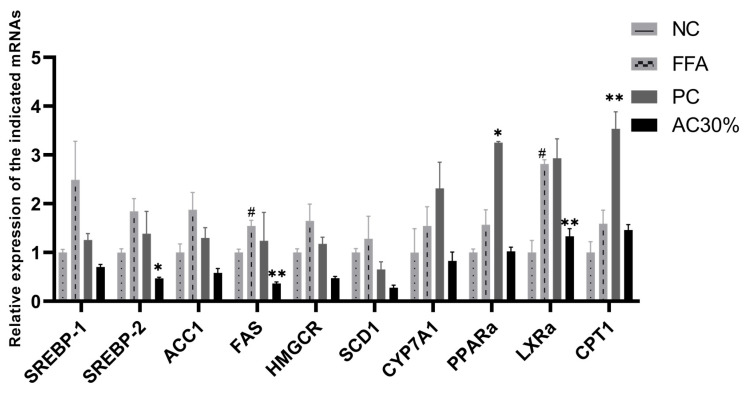
Relative mRNA expression of 10 genes after 30% ethanol treatment. Note: NC, the negative group; FFA, the model group; PC, the positive group; AC30%, the 30% ethanol fraction of ACE through D101 macroporous adsorption resin; * *p* < 0.05, ** *p* < 0.01 as compared to the model group of FFA-induced cells; # *p* < 0.05 as compared to the negative controls.

**Figure 9 antioxidants-12-01448-f009:**
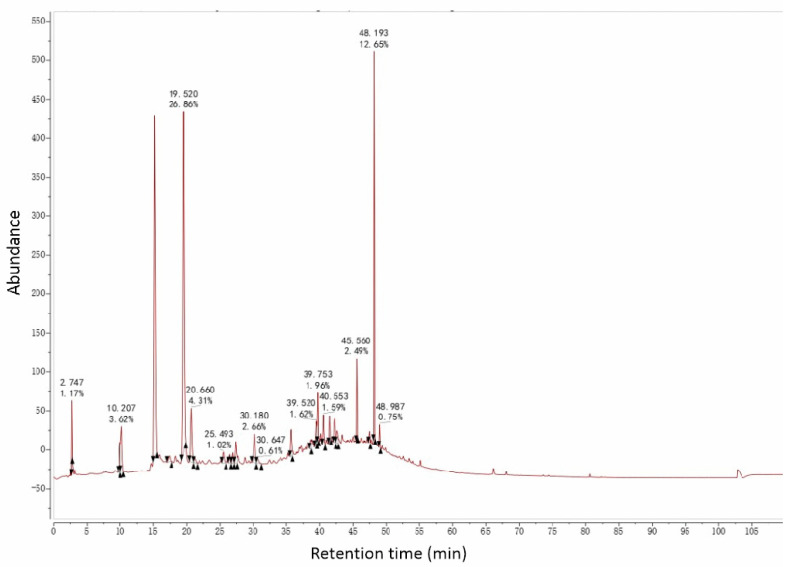
AC30% liquid chromatogram.

**Figure 10 antioxidants-12-01448-f010:**
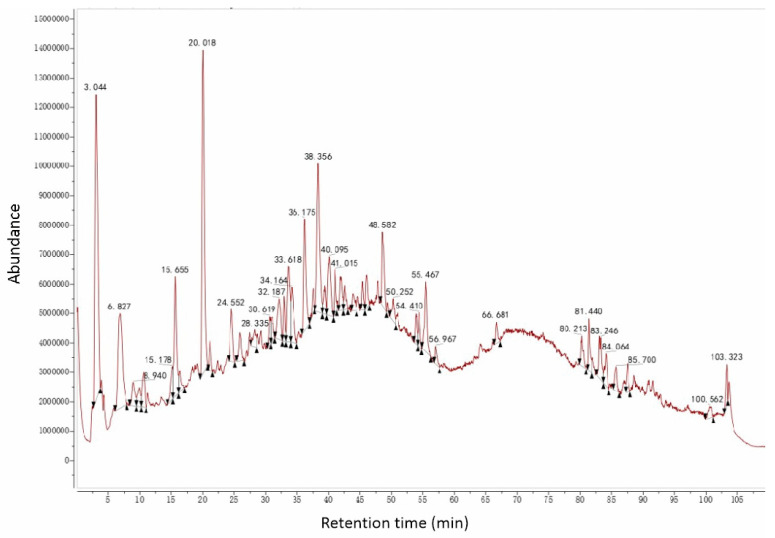
AC30% total ion current chromatogram.

**Figure 11 antioxidants-12-01448-f011:**
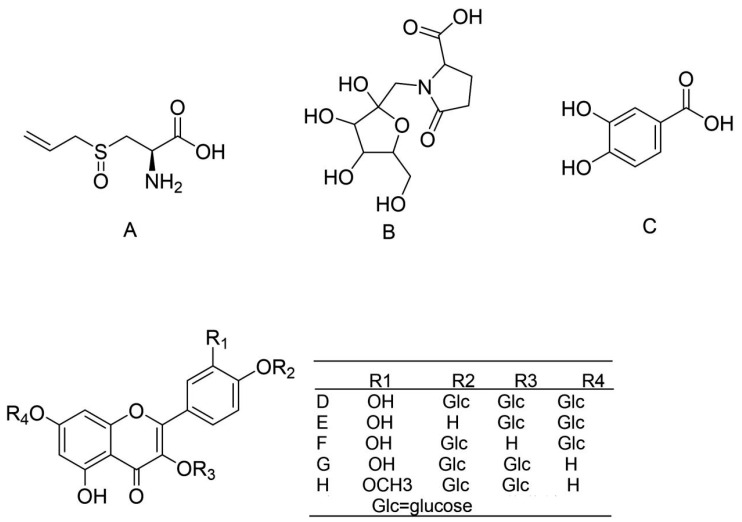
Compounds A-H from AC30%.

**Figure 12 antioxidants-12-01448-f012:**
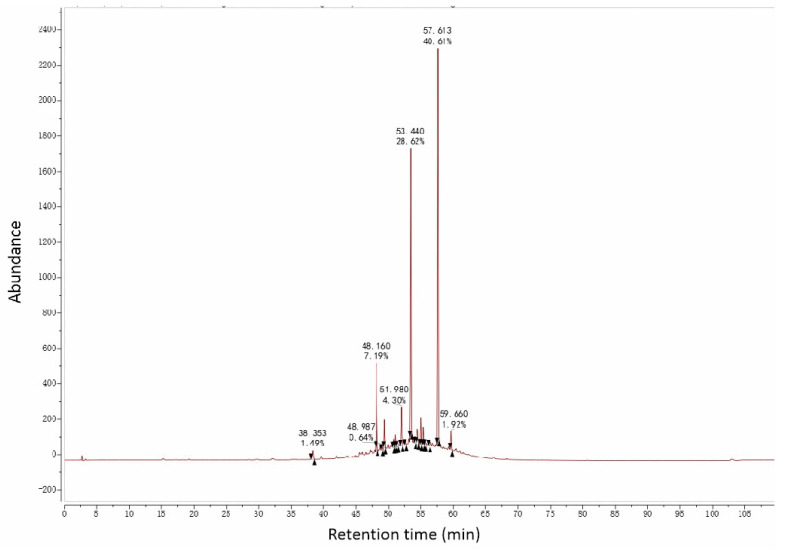
Liquid chromatogram of AC50%.

**Figure 13 antioxidants-12-01448-f013:**
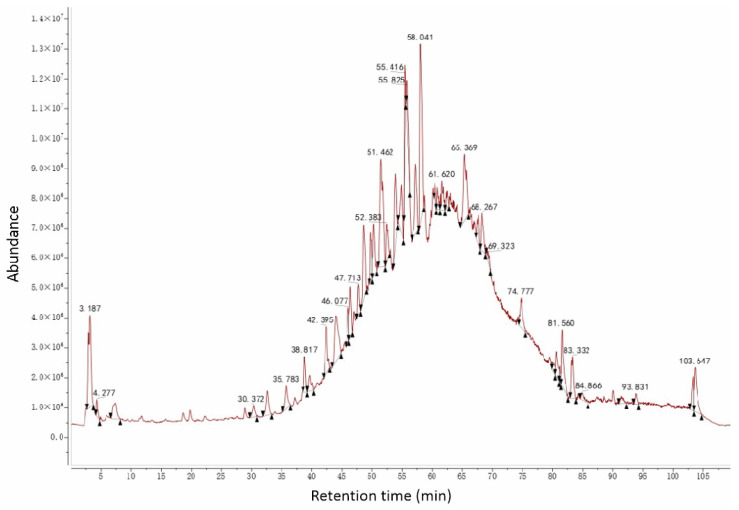
Total ion current chromatogram of AC50%.

**Figure 14 antioxidants-12-01448-f014:**
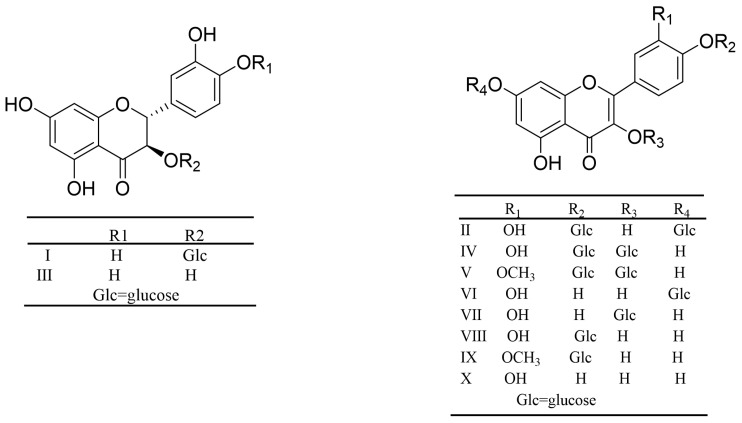
Compounds I-X from AC50%.

**Figure 15 antioxidants-12-01448-f015:**
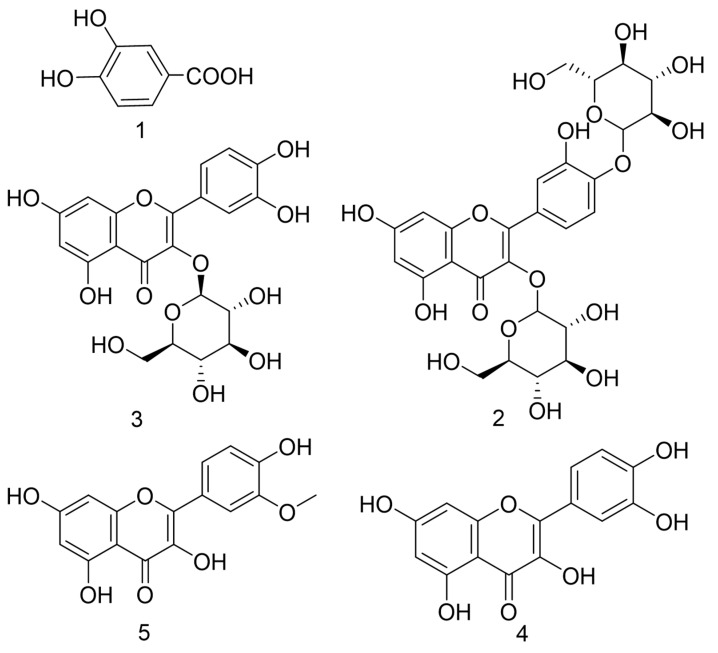
Compounds **1**–**5** from *Allium cepa*.

**Figure 16 antioxidants-12-01448-f016:**
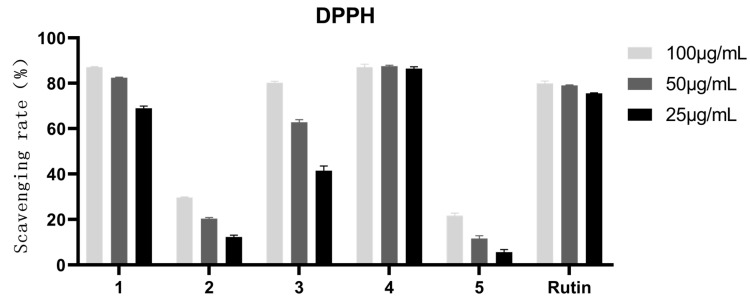
DPPH^●^ scavenging rate of the components from *A. cepa*. Note: Compound **1**, gallic acid; Compound **2**, quercetin-3,4-diglucoside; Compound **3**, Isoquercetin; Compound **4**, quercetin; Compound **5**, Isorhamnetin.

**Figure 17 antioxidants-12-01448-f017:**
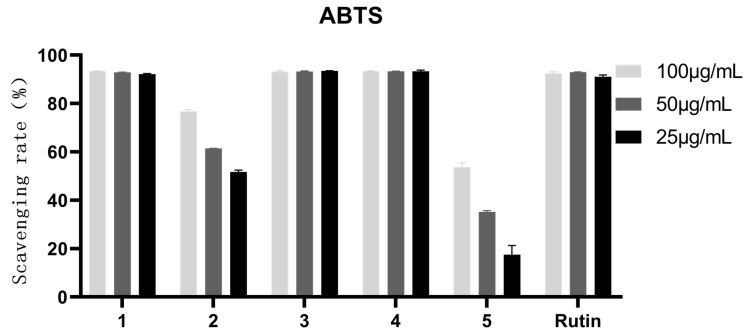
ABTS^+●^ scavenging rate of the components from *A. cepa*. Note: Compound **1**, gallic acid; Compound **2**, quercetin-3,4-diglucoside; Compound **3**, Isoquercetin; Compound **4**, quercetin; Compound **5**, Isorhamnetin.

**Figure 18 antioxidants-12-01448-f018:**
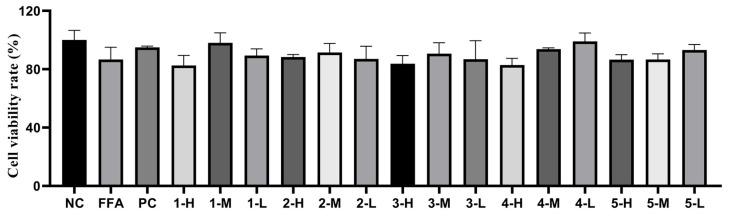
Results of cell survival test of the compounds from *Allium cepa*. Note: NC, the negative group; FFA, the model group; PC, thepositive group; Compound **1**, gallic acid; Compound **2**, quercetin-3,4’-diglucoside; Compound **3**, Isoquercetin; Compound **4**, quercetin; Compound **5**, Isorhamnetin.

**Figure 19 antioxidants-12-01448-f019:**
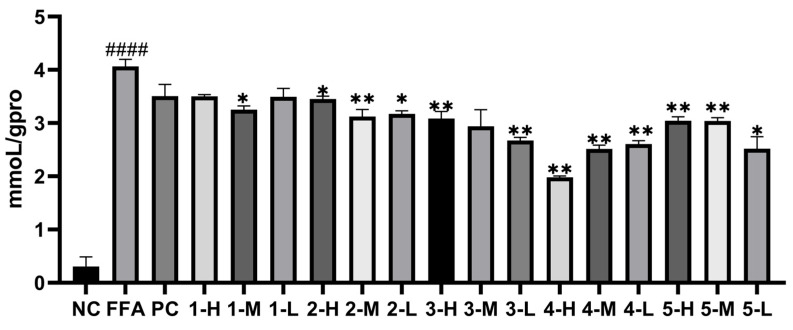
TG levels of the monomers against HepG2 cells after 24 h. Note: NC, the negative group; FFA, the model group; PC, the positive group; Compound **1**, gallic acid; Compound **2**, quercetin-3,4-diglucoside; Compound **3**, Isoquercetin; Compound **4**, quercetin; Compound **5**, Isorhamnetin, * *p* < 0.05, ** *p* < 0.01 as compared to the model group of FFA-induced cells; #### *p* < 0.0001 as compared to the negative controls.

**Figure 20 antioxidants-12-01448-f020:**
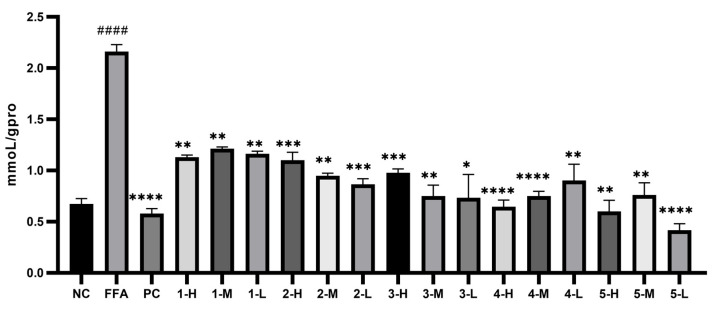
TC levels of the monomers against HepG2 cells after 24 h. Note: NC, the negative group; FFA, the model group; PC, the positive group; Compound **1**, gallic acid; Compound **2**, quercetin-3,4-diglucoside; Compound **3**, Isoquercetin; Compound **4**, quercetin; Compound **5**, Isorhamnetin, * *p* < 0.05, ** *p* < 0.01, *** *p* < 0.001, and **** *p* < 0.0001 as compared to the model group of FFA-induced cells; #### *p* < 0.0001 as compared to the negative controls.

**Figure 21 antioxidants-12-01448-f021:**
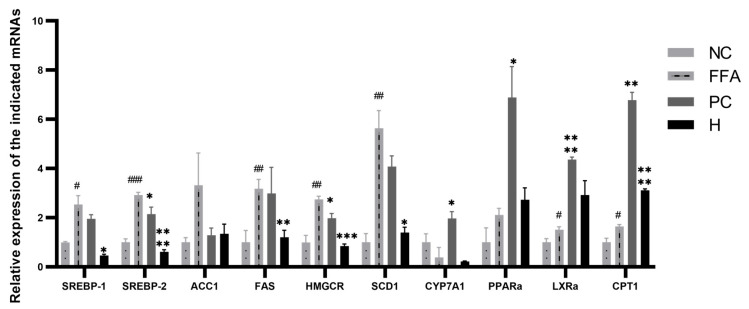
Relative mRNA expression of 10 genes after quercetin treatment. Note: NC, the negative group; FFA, the model group; PC, the positive group; H, quercetin, * *p* < 0.05, ** *p* < 0.01, *** *p* < 0.001 and as compared to the model group of FFA-induced cells; # *p* < 0.05, ## *p* < 0.01, and ### *p* < 0.001 as compared to the negative controls.

**Table 1 antioxidants-12-01448-t001:** Primer sequence table.

Genes	Gene Accession Number	Primer Sequence
h-*β*-actin	NM_001101	Forward: 5′-CCTAGAAGCATTTGCGGTGG-3′
		Reverse: 5′-GAGCTACGAGCTGCCTGACG-3′
h-SREBP-1	XM_054316990	Forward: 5′-CACCGTTTCTTCGTGGATGG-3′
		Reverse: 5′-CACACAGTTCAGTGCTCGCTCTA-3′
h-SREBP-2	XM_054325874	Forward: 5′-GCAGTCTGGTGGACAATGAGG-3′
		Reverse: 5′-TCATCCAATAGAGGGCTTCC-3′
h-FAS	XM_054315477	Forward: 5′-ACAGGGACAACCTGGAGTTCT-3′
		Reverse: 5′-CTGTGGTCCCACTTGATGAGT-3′
h-HMGCR	XM_054352485	Forward: 5′-AGGACCCCTTTGCTTAGATGA-3′
		Reverse: 5′-GCACCTCCACCAAGACCTATT-3′
h-ACC1	XM_054315912	Forward: 5′-ATTTCTTCCATCTCCCCCTCT-3′
		Reverse: 5′-ATGCCAATCTCATTTCCTCCT-3′
h-SCD1	NM_005063	Forward: 5′-TGGTTTCACTTGGAGCTGTG-3′
		Reverse: 5′-GGCCTTGGAGACTTTCTTCC-3′
h-CYP7A1	NM_000780	Forward: 5′-GTCTTTCCAGCCCTGGTAGC-3′
		Reverse: 5′-GAGGACCACGAGGTGTGTCT-3′
h-CPT1	XM_054367696	Forward: 5′-CGTCTTTTGGGATCCACGATT-3′
		Reverse: 5′-TGTGCTGGATGGTGTCTGTCTC-3′
h-LXRα	XM_054367384	Forward: 5′-AGCGCTTTGCCCACTTCA-3′
		Reverse: 5′-AGCCGGGTAGCTGTTTAGCA-3′
h-PPARα	XM_054325750	Forward: 5′-GACGTGCTTCCTGCTTCATAG-3′
		Reverse: 5′-CCACCATCGCGACCAGAT-3′

**Table 2 antioxidants-12-01448-t002:** Gradient elution conditions.

Time(min)	Phase A (%)	Phase B (%)
0	100	0
30	85	15
50	50	50
80	0	100
100	0	100
101	100	0
110	100	0

**Table 3 antioxidants-12-01448-t003:** Contents of total polyphenols and flavonoids in different fractions of onion.

Source	Fractions	TPC (mg GAE/g DW)	TFC (mg RE/g DW)
UAE	ACE	18.02	30.89
D101 resin	ACA	10.74	15.92
	AC30%	4.43	16.62
	AC50%	1.60	3.82
	AC70%	0.12	0.36
	AC95%	0.10	0.66

Note: ACE, the ethanol extract of *Allium cepa*; ACA, the water fraction of ACE through D101 macroporous adsorption resin column chromatography; AC30%, the 30% ethanol fraction of ACE through D101 macroporous adsorption resin column chromatography; AC50%, the 50% ethanol fraction of ACE through D101 macroporous adsorption resin column chromatography; AC70%, the 70% ethanol fraction of ACE through D101 macroporous adsorption resin column chromatography; AC95%, the 95% ethanol fraction of ACE through D101 macroporous adsorption resin column chromatography. mg GAE/g DW means the miligrams of gallic acid per grams of dried weight of *Allium cepa* powder. mg RE/g DW means the miligrams of rutin per grams of dried weight of *Allium cepa* powder.

**Table 4 antioxidants-12-01448-t004:** IC_50_.values of the fractions from *A. cepa* against ABTS^+●^ and DPPH^●^ free radicals.

Sample	DPPH^●^(μg/mL)	ABTS^+●^(μg/mL)
AC30%	-	24.15
AC50%	82.20	17.45
AC70%	90.03	18.52
Rutin	13.10	8.748

Note: AC30%, the 30% ethanol fraction of ACE through D101 macroporous adsorption resin column chromatography; AC50%, the 50% ethanol fraction of ACE through D101 macroporous adsorption resin column chromatography; AC70%, the 70% ethanol fraction of ACE through D101 macroporous adsorption resin column chromatography.

**Table 5 antioxidants-12-01448-t005:** Analysis results of AC30% using UPLC-ESI-MS/MS.

No.	t_R_ (min)	Identification	Composition	[M-H]^−^ (*m*/*z*)	MS/MS (*m*/*z*)	Error (ppm)	References
A	3.01	Alliin	C_6_H_11_NO_3_S	176.0404	145,120	5.727	Massbank Database, [28]
B	6.93	N-Fructosyl pyroglutamate	C_11_H_17_NO_8_	290.0884	128	4.764	[36]
C	20.02	Protocatechuic acid	C_7_H_6_O_4_	153.0209	109	7.097	[34]
D	40.16	Quercetin-3,7,4′-*O*-triglucoside	C_33_H_40_O_22_	787.1946	625,463,301	2.390	[35]
E	43.13	Quercetin-3,7-*O*-diglucoside	C_27_H_30_O_17_	625.1420	463,301	3.334	[35]
F	46.06	Quercetin-7,4′-*O*-diglucoside	C_27_H_30_O_17_	625.1420	463,301	3.334	[35]
G	48.58	Quercetin-3,4′-*O*-diglucoside	C_27_H_30_O_17_	625.1420	463,301	3.334	[35]
H	49.40	Isorhamnetin-3,4′-*O*-diglucoside	C_28_H_32_O_17_	639.1579	477, 315	3.679	[35]

**Table 6 antioxidants-12-01448-t006:** Analysis results of AC50% using UPLC-ESI-MS/MS.

No.	t_R_ (min)	Identification	Composition	[M-H]^−^ (*m*/*z*)	MS/MS (*m*/*z*)	Error (ppm)	References
Ⅰ	45.09	Taxifolin-3-glucoside	C_21_H_22_O_12_	465.1045	285, 151	0.990	[9], PubChem
Ⅱ	46.08	Quercetin-7,4′-diglucoside	C_27_H_30_O_17_	625.1422	463, 301	3.595	[35]
Ⅲ	47.68	Taxifolin	C_15_H_12_O_7_	303.0519	151, 125	0.980	[9], PubChem
Ⅳ	48.57	Quercetin-3,4′-diglucoside	C_27_H_30_O_17_	625.1422	463, 301	3.595	[35]
Ⅴ	49.38	Isorhamnetin-3,4′-diglucoside	C_28_H_32_O_17_	639.1581	477, 315	4.012	[35]
Ⅵ	49.79	Quercetin-7-*O*-glucoside	C_21_H_20_O_12_	463.0892	301	4.474	[35]
Ⅶ	52.38	Quercetin-3-*O*-glucoside	C_21_H_20_O_12_	463.0892	300	4.474	[35]
Ⅷ	53.92	Quercetin-4′-*O*-glucoside	C_21_H_20_O_12_	463.0892	301	4.474	[35]
Ⅸ	55.42	Isorhamnetin-4′-glucoside	C_22_H_22_O_12_	477.1045	315	3.723	[35]
Ⅹ	58.04	Quercetin	C_15_H_10_O_7_	301.037	151,107	1.000	[35]

**Table 7 antioxidants-12-01448-t007:** IC_50_ of the components from *A. cepa* against ABTS**^+^^●^** and DPPH**^●^** free radicals.

Sample	DPPH^●^(μg/mL)	ABTS^+●^(μg/mL)
Rutin	13.10	8.748
1	12.76	2.865
2	-	17.87
3	31.36	5.484
4	7.12	2.121
5	-	96.83

Note: Compound **1**, gallic acid; Compound **2**, quercetin-3,4’-diglucoside; Compound **3**, Isoquercetin; Compound **4**, quercetin; Compound **5**, Isorhamnetin.

## Data Availability

The original contributions presented in this study are included in the article. Further inquiries can be directed to the corresponding author.

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
