# Peer review of "Bio-Assay-Guided Isolation of Fractions and Constituents with Antioxidant and Lipid-lowering Activity from Allium cepa"

_antioxidants, 2023, doi:10.3390/antiox12071448_

Round 1

Reviewer 1 Report

This is a well-designed study which supports novel and interestings findings. Minor revision is required before considering for acceptance.

- The abstract is too long and It should be a little bit shorten

- Statistical analysis section is poor. The authors should added more detailed information about the statistical methods used and thry should the software which was used for statistical analysis.

- A discussion section is strongly recommended to be added separately from results section.

- In the discussion section the authors should compared their results with previous published data.

 - In the discussion section the authors should expllain whether their results are complementary in - respect to the previous published data and whether their results cover a literature gap, emphasizing the importance od their findings.

- At the end of the discusion section a separate paragraphs reporting the strengths and the limitations of the study should be reported.

- In the conclusion section, the author should add some statements about the future studies that could be performed in this topic.

The reference list is poor and should be enhriced with several stydies related with the topic, e.g.

·       Clin Exp Pharmacol Physiol. 2023 Mar;50(3):205-217. doi: 10.1111/1440-1681.13743.

·       Curr Med Chem. 2023 May 17. doi: 10.2174/0929867330666230517124033.

·       Molecules. 2022 Feb 24;27(5):1533. doi: 10.3390/molecules27051533.

·       Phytother Res. 2021 Oct;35(10):5647-5667. doi: 10.1002/ptr.7202.

·       Expert Rev Anti Infect Ther. 2021 Apr;19(4):519-528. doi: 10.1080/14787210.2021.1828061.

·       Oxid Med Cell Longev. 2020 Feb 13;2020:1356893. doi: 10.1155/2020/1356893.

·       Plant Physiol Biochem. 2019 Nov;144:135-143. doi: 10.1016/j.plaphy.2019.09.039.

·       Am J Cardiovasc Dis. 2017 Apr 15;7(2):19-32.

·       Curr Top Med Chem. 2017;17(12):1336-1370. doi: 10.2174/1568026617666170102125648.

·       J Res Med Sci. 2015 May;20(5):491-502. doi: 10.4103/1735-1995.163977.

The above reference coyld be added eithe in the introduction section or in the discussion section or in both of the,

The quality of English language is quitely adequate. Minor English language editing is recommended.

Author Response

To Reviewer 1

This is a well-designed study which supports novel and interesting findings. Minor revision is required before considering for acceptance.

1.Reviewerscomment:The abstract is too long and It should be a little bit shorten

Authors’ reply: The abstract has been revised to be a total of about 200 words (Page 1, Line 13-29).

2.Reviewers comment: Statistical analysis section is poor. The authors should added detailed information about the statistical methods used.

Authors’ reply: In statistical analysis section, detailed information about the statistical methods used have been added (Page 9, Line 323-332).

3.Reviewers comment: A discussion section is strongly recommended to be added separately from results section.

Authors’ reply: According to the Instructions for Authors of Antioxidants, the discussion section may be combined with results section. Moreover, we just have 5 days to revise this manuscript. Considering the above conditions, we still combine the results section and the discussion section together.

4.Reviewers comment: In the discussion section the authors should compared their results with previous published data.

Authors’ reply: In the discussion section, we have added the discussions compared the results with previous published data (Page 9, Line 353-354, 374-376; Page 12, Line 450-451; Page 15, Line 514-517; Page 17, Line 536-538; Page 20, Line 582-584).

5.Reviewers comment: In the discussion section the authors should explain whether their results are complementary in respect to the previous published data and whether their results cover a literature gap, emphasizing the importance or their findings.

Authors’ reply: As for the discussion section, both comparing the results with previous published data and explaining whether their results are complementary in respect to the previous published data have been done (Page 9, Line 354-355, 374-376; Page 12, Line 452-453; Page 20, Line 584-586).

6.Reviewers comment:The strengths and the limitations should be reported. 

Authors’ reply: The strengths and the limitations have been reported (Page 23, Line 638-639,663-664).

7. Reviewerscomment: In the conclusion section, the author should add some statements about the future studies that could be performed in this topic.

Authors’ reply: In the conclusion section, some statements about the future studies that could be performed in this topic have been added (Page 23, Line 663-664).

8. Reviewers comment:The reference list is poor and should be enhriced with several stydies related with the topic, e.g.

  • Clin Exp Pharmacol Physiol. 2023 Mar;50(3):205-217. doi: 10.1111/1440-1681.13743.
  • Curr Med Chem. 2023 May 17. doi: 10.2174/0929867330666230517124033.
  • 2022 Feb 24;27(5):1533. doi: 10.3390/molecules27051533.
  • Phytother Res. 2021 Oct;35(10):5647-5667. doi: 10.1002/ptr.7202.
  • Expert Rev Anti Infect Ther. 2021 Apr;19(4):519-528. doi: 10.1080/14787210.2021.1828061.
  • Oxid Med Cell Longev. 2020 Feb 13;2020:1356893. doi: 10.1155/2020/1356893.
  • Plant Physiol Biochem. 2019 Nov;144:135-143. doi: 10.1016/j.plaphy.2019.09.039.
  • Am J Cardiovasc Dis. 2017 Apr 15;7(2):19-32.
  • Curr Top Med Chem. 2017;17(12):1336-1370. doi: 10.2174/1568026617666170102125648.
  • J Res Med Sci. 2015 May;20(5):491-502. doi: 10.4103/1735-1995.163977.

The above reference coyld be added eithe in the introduction section or in the discussion section or in both of the,

Authors’ reply: This article (Plant Physiol Biochem. 2019 Nov;144:135-143. doi: 10.1016/j.plaphy.2019.09.039) has been cited (Page 20, 584; Page 26, 804-805).

Thank you for your attentions and thanks a lot the referees who have done a great job on our manuscript.

With best regards!

Dr. Chunyan Yang

School of Pharmacy, Institute of Material Medica, North Sichuan Medical College

Nanchong 637100, P. R. China

E-mail: yangchy2023@nsmc.edu.cn 

Reviewer 2 Report

I have reviewed the manuscript entitled “Bio-assay Guided Isolation of the Fractions and Constituents with Antioxidant and Lipid-lowering Activity from Allium cepa" and found it to be well-written, with many positive aspects that make it suitable for publication in the journal "Antioxidants".

This original article presents the bio-assay guided isolation of fractions and constituents from Allium cepa with antioxidant and lipid-lowering activity. The authors employed a systematic approach to isolate and characterize the active compounds and evaluated their potential therapeutic benefits. The study sheds light on the pharmacological properties of Allium cepa and its potential application in combating oxidative stress and lipid-related disorders.

I have no substantive comments on the work and propose that the editorial team accept the manuscript after minor revisions.

Here are my minor comments and required corrections:

Abstract: The abstract should be a total of about 200 words maximum. Please, reduce the words.

Expand the introduction to clearly outline the current challenges in combating oxidative stress and lipid-related disorders, and highlight the potential of natural compounds from Allium cepa as effective therapeutic agents.

Strengthen the discussion section by thoroughly analyzing the obtained results, including a comparison with previous studies on Allium cepa extracts, and providing potential mechanisms of action for the observed antioxidant and lipid-lowering effects.

The conclusions section of the manuscript provides a concise summary of the findings and their implications. It effectively highlights the potential of the isolated fractions and constituents from Allium cepa as promising sources of antioxidants and lipid-lowering agents. However, it would be beneficial to further elaborate on the significance of these findings in the broader context of oxidative stress-related disorders and lipid metabolism. By discussing the potential applications and future directions of the research, the conclusions section can be strengthened to provide a more comprehensive and forward-looking perspective.

By incorporating these suggested improvements, the manuscript will offer a more comprehensive and well-structured account of the bio-assay guided isolation of fractions and constituents with antioxidant and lipid-lowering activity from Allium cepa. This will enhance its suitability for publication in the journal "Antioxidants".

Author Response

To Reviewer 2

I have reviewed the manuscript entitled “Bio-assay Guided Isolation of the Fractions and Constituents with Antioxidant and Lipid-lowering Activity from Allium cepa" and found it to be well-written, with many positive aspects that make it suitable for publication in the journal "Antioxidants".

This original article presents the bio-assay guided isolation of fractions and constituents from Allium cepa with antioxidant and lipid-lowering activity. The authors employed a systematic approach to isolate and characterize the active compounds and evaluated their potential therapeutic benefits. The study sheds light on the pharmacological properties of Allium cepa and its potential application in combating oxidative stress and lipid-related disorders.

I have no substantive comments on the work and propose that the editorial team accept the manuscript after minor revisions.

1. Reviewerscomment:Abstract: The abstract should be a total of about 200 words maximum. Please, reduce the words.

Authors’ reply: The abstract has been revised to be a total of about 200 words (Page 1, Line 13-29).  

2.Reviewers comment: Expand the introduction to clearly outline the current challenges in combating oxidative stress and lipid-related disorders, and highlight the potential of natural compounds from Allium cepa as effective therapeutic agents.

Authors’ reply: Expanding the introduction to clearly outline the current challenges in combating oxidative stress and lipid-related disorders, and highlighting the potential of natural compounds from Allium cepa as effective therapeutic agents have been done (Page 1, Line 34-45; Page 2, Line 46-83).

3.Reviewers comment: Strengthen the discussion section by thoroughly analyzing the obtained results, including a comparison with previous studies on Allium cepa extracts, and providing potential mechanisms of action for the observed antioxidant and lipid-lowering effects. 

Authors’ reply: Strengthening the discussion section by thoroughly analyzing the obtained results, including a comparison with previous studies on Allium cepa extracts have been done (Page 9, Line 353-354, 374-376; Page 12, Line 450-451; Page 15, Line 514-517; Page 17, Line 536-538; Page 20, Line 582-584), and providing potential mechanisms of action for the observed antioxidant and lipid-lowering effects have been done (Page 20, Line 584-586).

4.Reviewers comment:The conclusions section of the manuscript provides a concise summary of the findings and their implications. It effectively highlights the potential of the isolated fractions and constituents from Allium cepa as promising sources of antioxidants and lipid-lowering agents. However, it would be beneficial to further elaborate on the significance of these findings in the broader context of oxidative stress-related disorders and lipid metabolism. By discussing the potential applications and future directions of the research, the conclusions section can be strengthened to provide a more comprehensive and forward-looking perspective. 

Authors’ reply: In the conclusions section of the manuscript, the significance of these findings in the broader context of oxidative stress-related disorders and lipid metabolism has been elaborated (Page 23, 663-665). In the conclusions section, the potential applications and future directions of the research have been added (Page 23, 662-663).

Thank you for your attentions and thanks a lot the referees who have done a great job on our manuscript.

With best regards!

Dr. Chunyan Yang

School of Pharmacy, Institute of Material Medica, North Sichuan Medical College

Nanchong 637100, P. R. China

E-mail: yangchy2023@nsmc.edu.cn